# Efficient Dataset Distillation
# using Random Feature Approximation

**Noel Loo, Ramin Hasani, Alexander Amini, Daniela Rus**
Computer Science and Artificial Intelligence Lab (CSAIL)
Massachusetts Institute of Technology (MIT)
{loo, rhasani, amini, rus} @mit.edu

## Abstract

Dataset distillation compresses large datasets into smaller synthetic coresets which retain performance with the aim of reducing the storage and computational burden of processing the entire dataset. Today's best-performing algorithm, *Kernel Inducing Points* (KIP), which makes use of the correspondence between infinite-width neural networks and kernel-ridge regression, is prohibitively slow due to the exact computation of the neural tangent kernel matrix, scaling $O(|S|^2)$, with $|S|$ being the coreset size. To improve this, we propose a novel algorithm that uses a random feature approximation (RFA) of the Neural Network Gaussian Process (NNGP) kernel, which reduces the kernel matrix computation to $O(|S|)$. Our algorithm provides at least a 100-fold speedup over KIP and can run on a single GPU. Our new method, termed an RFA Distillation (RFAD), performs competitively with KIP and other dataset condensation algorithms in accuracy over a range of large-scale datasets, both in kernel regression and finite-width network training. We demonstrate the effectiveness of our approach on tasks involving model interpretability and privacy preservation.[1]

## 1  Introduction

Coreset algorithms aim to summarize large datasets into significantly smaller datasets that still accurately represent the full dataset on downstream tasks [Jubran et al., 2019]. There are myriad applications of these smaller datasets including speeding up model training [Mirzasoleiman et al., 2020], reducing catastrophic forgetting [Aljundi et al., 2019, Rebuffi et al., 2017, Borsos et al., 2020], and enhancing interpretability [Kim et al., 2016, Bien and Tibshirani, 2011]. While most coreset selection techniques aim to select representative data points from the dataset, recent work has looked at generating synthetic data points instead, a process known as dataset distillation [Wang et al., 2018, Bohdal et al.,

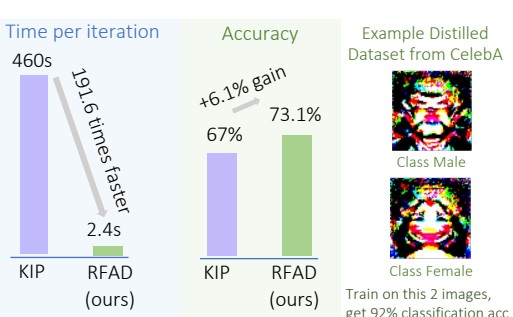

Figure 1: RFAD provides over 100-fold speedup over the state-of-the-art algorithm Kernel-Inducing Points (KIP) [Nguyen et al., 2021a], while exceeding its performance on CIFAR-10. (right) Example distilled synthetic sets one image per class

2020, Sucholutsky and Schonlau, 2019, Zhao et al., 2021, Zhao and Bilen, 2021b, Nguyen et al., 2021b]. These synthetic datasets have the benefit of using continuous gradient-based optimization techniques rather than combinatorial methods and are not limited to the set of images and labels given by the dataset, providing added flexibility and performance.

---

[1]Code is available at `https://github.com/yolky/RFAD`

A large variety of applications benefit from obtaining an efficient dataset distillation algorithm. For instance, Kernel methods [Vinyals et al., 2016, Kaya and Bilge, 2019, Snell et al., 2017, Ghorbani et al., 2020, Refinetti et al., 2021] usually demand a large support set in order to generate good prediction performance at inference. This can be facilitated by an efficient dataset distillation pipeline. Moreover, distilling a synthetic version of sensitive data helps preserve privacy; a support set can be provided to an end-user for the downstream applications without disclosure of data. Lastly, for resource-hungry applications such as continual learning [Borsos et al., 2020], neural architecture search [Shleifer and Prokop, 2019] and automated machine learning [Hutter et al., 2019], generation of a support-set on which we can fit models efficiently is very helpful.

Recently, a dataset distillation method called Kernel-Inducing Points (KIP) [Nguyen et al., 2021a,b] showed great performance in neural network classification tasks. KIP uses Neural Tangent Kernel (NTK) ridge-regression to *exactly* compute the output states of an infinite-width neural network trained on the support set. Although the method established the state-of-the-art for dataset distillation in terms of accuracy, the computational complexity of KIP is very high due to the exact calculation of the NTK. The algorithm, therefore, has limited applicability.

In this paper, we build on the prior work on KIP and develop a new algorithm for dataset distillation called RFAD, which has similar accuracy and significantly better performance than KIP. The key insight is to introduce a new kernel inducing point method that improves complexity from $O(|S|^2)$ (where $|S|$ is the support-set size) to $O(|S|)$. To this end, we make three **major contributions**:

**I.** We develop RFAD, a fast, accurate, and scalable algorithm for dataset distillation in neural network classification tasks.

**II.** We improve the time performance of KIP [Nguyen et al., 2021a,b] by over two orders of magnitude while retaining or improving its accuracy. This speedup comes from leveraging a random-feature approximation of the Neural Network Gaussian Process (NNGP) kernel by instantiating random neural networks.

**III.** We show the effectiveness of RFAD in efficient dataset distillation tasks, enhancing model interpretability and privacy preservation.

## 2 Background and Related Work

**Coresets and Dataset Distillation.** Coresets are a subset of data that ensure models trained on them show competitive performance compared to models trained directly on data. Standard coreset selection algorithms use importance sampling to find coresets [Har-Peled and Mazumdar, 2004, Lucic et al., 2017, Cohen et al., 2017]. Besides random selection methods, inspired by catastrophic forgetting [Toneva et al., 2019] and mean-matching (adding samples to the coreset to match the mean of the original dataset) [Chen et al., 2010, Rebuffi et al., 2017, Castro et al., 2018, Belouadah and Popescu, 2020, Scholkopf et al., 1999], new algorithms have been introduced. An overview of how coresets work on point approximation is provided in Phillips [2016]

More recently, aligned with coreset selection methods, new algorithms have been developed to distill a synthetic dataset from a given dataset, such that fitting to this synthetic set provides performance on par with training on the original dataset [Wang et al., 2018]. To this end, these dataset condensation (or distillation) algorithms use gradient matching [Zhao et al., 2021, Maclaurin et al., 2015, Lorraine et al., 2020], utilize differentiable siamese augmentation [Zhao and Bilen, 2021b], and matching distributions [Zhao and Bilen, 2021a]. Dataset distillation has also been applied to the labels rather than images [Bohdal et al., 2020]. Recently, a novel algorithm called Kernel Inducing Points (KIP) [Nguyen et al., 2021a,b] has been introduced that performs very well on distilling synthetic sets by using neural tangent kernel ridge-regression (KRR). KIP, similar to other algorithms, is computationally expensive. Here, we propose a new method to significantly improve its complexity.

**Infinite Width Neural Networks.** Single-layer infinite-width randomly initialized neural networks correspond to Gaussian Processes [Neal, 1996], allowing for closed-form exact training of Bayesian neural networks for regression. Recently, this has been extended to deep fully-connected networks [Lee et al., 2018, de G. Matthews et al., 2018], convolutional networks [Novak et al., 2019, Garriga-Alonso et al., 2019], attention-based networks [Hron et al., 2020], and even to arbitrary neural architectures [Yang, 2019], with the corresponding GP kernel being the NNGP Kernel. Likewise, for infinite-width neural networks trained with gradient descent, the training process simplifies dramati-

cally, corresponding to kernel ridge regression when trained with MSE loss with the corresponding kernel being the Neural Tangent Kernel (NTK) [Jacot et al., 2018, Arora et al., 2019, Loo et al., 2022]. These two kernels are closely related, as the NNGP kernel forms the leading term of the NTK kernel, representing the effect of the final layer weights. Calculation of kernel entries typically scales with $O(HWD)$ for conv nets, with $H, W$ being the image height and width, $D$ the network depth, and $O(H^2W^2D)$ for architectures with global average pooling [Arora et al., 2019]. This, combined with the necessity of computing and inverting the $N \times N$ kernel matrix for kernel ridge regression, typically make these methods intractable for large datasets [Snelson and Ghahramani, 2006, Titsias, 2009].

**Random Feature methods.** Every kernel corresponds to a dot product for some feature map: $k(x, x') = \phi(x)^T \phi(x')$. Random feature methods aim to approximate the feature vector with a finite-dimensional random feature vector, the most notable example being Random Fourier Features [Rahimi and Recht, 2007]. Typically, this limits the rank of the kernel matrix, enabling faster matrix inversion and allowing for scaling kernel methods to large datasets. Recently, these random feature methods have been used to speed up NTKs and NNGPs [Zandieh et al., 2021, Novak et al., 2022, 2019] at inference or for neural architecture search [Peng et al., 2020]. In this work, we focus on the NNGP approximation described in Novak et al. [2019], as it only requires network forward passes and is model agnostic, allowing for flexible usage across different architectures without more complex machinery needed to calculate the approximation, unlike those found in Zandieh et al. [2021], Novak et al. [2022].

## 3 Algorithm Setup and Design

In this section, we first provide a high-level background on the KIP algorithm. We then sequentially outline our modifications leading to the RFAD algorithm.

### 3.1 KIP Revisit

The Kernel-Inducing Point algorithm [Nguyen et al., 2021a,b], or KIP, is a dataset distillation technique that uses the NTK kernel ridge-regression correspondence to compute exactly the outputs of an infinite-width neural network trained on the support set, bypassing the need to ever compute gradients or back-propagate on any finite network. Let $X_T, y_T$ correspond to the images and one-hot vector labels on the training dataset and let $X_S, y_S$ be the corresponding images and labels for the support set, which we aim to optimize. We have the outputs of a trained neural network as $f(X_T) = K_{TS}(K_{SS} + \lambda I)^{-1}y_S$, with $K$ being the kernel matrices calculated using the NTK kernel, with $T \times S$ or $S \times S$ entries, for $K_{TS}$ and $K_{SS}$, respectively. $\lambda$ is a small regularization parameter. KIP then optimizes $L_{MSE} = ||y_T - f(X_T)||_2^2$ directly. The key bottleneck is the computation of these kernel matrices, requiring $O(TS \cdot HWD)$ time and memory, necessitating the use of hundreds of GPUs working in parallel. Additionally, the use of the MSE loss is suboptimal.

### 3.2 Replacing the NTK Kernel with an NNGP Kernel

We first replace the NTK used in the kernel regression of KIP with an NNGP kernel. While this change alone would yield a speed up, as the NNGP kernel is less computationally intensive to compute [Novak et al., 2020], we primarily do this because the NNGP kernel admits a simple random feature approximation, with advantages described later in this section. We first justify the appropriateness of this modification.

Firstly, we denote that in the computation of NTK ($\Theta$) and NNGP ($K$) forms the leading term, as shown in Table 1 in Appendix D of [Novak et al., 2020] which outlines the NTK and NNGP kernel computation rules for various layers of a neural network. For fully connected (FC) layers, which is the typical final layer in neural network architectures, the remaining terms are suppressed by a matrix of expected derivatives with respect to activations, $\dot{K}$, as observed by the recursion yielded from the computation of the NTK for an FC network [Novak et al., 2020]: $\Theta^l = K^l + \dot{K}^l \odot \Theta^{l-1}$. For ReLU activations, the entries in this derivative matrix are upper bounded by 1, so the remaining terms must have a decaying contribution. We verify that our algorithms still provide good performance under the NTK and for finite networks trained with gradient descent, justifying this approximation.

**Algorithm 1** Dataset distillation with NNGP random features

---

**Require:** Training set and labels $X_T, y_T$, Randomly initialized coreset and labels $X_S, y_S$, Random network count $N$, Random network output dimension $M$, Batch size $|B|$, Random network initialization distribution, $p(\theta)$, Regularization coefficient, $\lambda$, Learning rate $\eta$,

    **while** loss not converged **do**

        Sample batch from the training set $X_B, y_B \sim p(X_T, y_T)$

        Sample $N$ random networks each with output dimension $M$ from $p(\theta)$: $\theta_1, ...\theta_N \sim p(\theta)$

        Compute random features for batch with random nets:

$$\hat{\Phi}(X_B) \leftarrow \tfrac{1}{\sqrt{NM}}[f_{\theta_1}(X_B),...,f_{\theta_N}(X_B)]^T \in \mathbb{R}^{|NM| \times |B|}$$

        Compute random features for support set with random nets:

$$\hat{\Phi}(X_S) \leftarrow \tfrac{1}{\sqrt{NM}}[f_{\theta_1}(X_S),...,f_{\theta_N}(X_S)]^T \in \mathbb{R}^{|NM| \times |S|}$$

        Compute kernel matrices: $\hat{K}_{BS} \leftarrow \hat{\Phi}(X_B)^T \hat{\Phi}(X_S)$

    $\hat{K}_{SS} \leftarrow \hat{\Phi}(X_S)^T \hat{\Phi}(X_S)$

        Calculate trained network output on batch: $\hat{y}_B \leftarrow \hat{K}_{BS}(\hat{K}_{SS} + \lambda I_{|S|})^{-1} y_S$

        Calculate loss: $\mathcal{L} = \mathcal{L}(y_B, \hat{y}_B)$

        Update coreset: $X_S \leftarrow X_S - \eta \frac{\partial \mathcal{L}}{\partial X_S}, y_S \leftarrow y_S - \eta \frac{\partial \mathcal{L}}{\partial y_S}$

    **end while**

---

### 3.3 Replacing NNGP with an Empirical NNGP

When we sample from a Gaussian process $f \sim \mathcal{GP}(0, K)$, it suggests a natural finite feature map corresponding to scaled draws from the GP: $\hat{\phi}(x) = \frac{1}{\sqrt{N}}[f_1(x), ..., f_N(x)]^T$. For most GPs, this insight is not relevant, as sampling from a GP typically requires a Cholesky decomposition of the kernel matrix, requiring its computation in the first place [Rasmussen and Williams, 2006]. However, for NNGP we can generate approximate samples of $f$ by instantiating random neural networks, $f_i(x) = f_{\theta_i}(x), \theta_i \sim p(\theta)$, for some initialization distribution $p(\theta)$. Moreover, with a given neural network, we can define $f_i$ to be a vector of dimension $M$ by having a network with multiple output heads, meaning that with $N$ networks, we have $NM$ features. For our purposes, we typically have N = 8, M = 4096, giving 32768 total features. For the convolutional architectures we consider, this corresponds to $C = 256$ convolutional channels per layer. Even with this relatively large number of features, we still see a significant computation speedup over exact calculation.

To sample $f \sim \mathcal{GP}(0, K)$, we would have to instantiate random *infinite* width neural nets, whereas, in practice, we can only sample finite ones. This discrepancy incurs an $O(1/C)$ bias to our kernel matrix entries, with $C$ being the width-relevant parameter (i.e., convolutional channels) [Yaida, 2020]. However, we have a $O(1/(NC))$ variance of the mean of the random features [Daniely et al., 2016], meaning that in practice, the variance dominates the computation over bias. This has been noted empirically in Novak et al. [2019], and we verify that the finite-width bias does not significantly affect performance in appendix I, showing that we can achieve reasonable performance with as little as *one* convolution channel.

The time cost of computing these random features is linear in the training set and coreset size, $|T|, |S|$. With the relatively low cost of matrix multiplication, this results in the construction of the kernel matrices $K_{TS}$ and $K_{SS}$ having $O(|T| + |S|)$ and $O(|S|)$, time complexity, respectively, as opposed to $O(|T||S|)$ and $O(|S|^2)$ with KIP. Noting that the cost of matrix inversion is relatively small compared to random feature construction, our total runtime is reduced to **linear** in the coreset size. We empirically verify this linear time complexity in section 4.1 and additionally provide a more detailed discussion in appendix C.

### 3.4 Loss Function in dataset distillation

We denoted earlier that $L_{MSE}$ is not well suited for dataset distillation settings. In particular, there are two key problems:

**Over-influence of already correctly classified data points.** Consider two-way classification, with the label 1 corresponding to the positive class and $-1$ corresponding to the negative class. Let $x_1$ and $x_2$ be items in the training set whose labels are both 1. Let $f_{\text{KRR}}(x) = K_{x,S}(K_{SS} + \lambda I)^{-1} y_S$ be the KRR output on $x$ given our support set $X_S$. If $f_{\text{KRR}}(x_1) = 5$ and $f_{\text{KRR}}(x_2) = -1$, then the

Table 1: Kernel distillation results on five datasets with varying support set sizes. **Bolded** numbers indicate the best performance with fixed labels, and underlined numbers indicate the best performance with learned labels. Note that DC and DSA use fixed labels. (n = 4)

| | | Fixed Labels | | | | Learned Labels | |
|---|---|---|---|---|---|---|---|
| | Img/Cls | DC | DSA | KIP | RFAD (ours) | KIP | RFAD (ours) |
| MNIST | 1 | $91.7 \pm 0.5$ | $88.7 \pm 0.6$ | $95.2 \pm 0.2$ | $\mathbf{96.7 \pm 0.2}$ | $97.3 \pm 0.1$ | $97.2 \pm 0.2$ |
| | 10 | $97.4 \pm 0.2$ | $97.8 \pm 0.1$ | $98.4 \pm 0.0$ | $\mathbf{99.0 \pm 0.1}$ | $99.1 \pm 0.1$ | $99.1 \pm 0.0$ |
| | 50 | $98.8 \pm 0.1$ | $\mathbf{99.2 \pm 0.1}$ | $99.1 \pm 0.0$ | $99.1 \pm 0.0$ | $99.4 \pm 0.1$ | $99.1 \pm 0.0$ |
| Fashion-MNIST | 1 | $70.5 \pm 0.6$ | $70.6 \pm 0.6$ | $78.9 \pm 0.2$ | $\mathbf{81.6 \pm 0.6}$ | $82.9 \pm 0.2$ | $84.6 \pm 0.2$ |
| | 10 | $82.3 \pm 0.4$ | $84.6 \pm 0.3$ | $87.6 \pm 0.1$ | $\mathbf{90.0 \pm 0.1}$ | $91.0 \pm 0.1$ | $90.3 \pm 0.2$ |
| | 50 | $83.6 \pm 0.4$ | $88.7 \pm 0.2$ | $90.0 \pm 0.1$ | $\mathbf{91.3 \pm 0.1}$ | $92.4 \pm 0.1$ | $91.4 \pm 0.1$ |
| SVHN | 1 | $31.2 \pm 1.4$ | $27.5 \pm 1.4$ | $48.1 \pm 0.7$ | $\mathbf{51.4 \pm 1.3}$ | $64.3 \pm 0.4$ | $57.4 \pm .8$ |
| | 10 | $76.1 \pm 0.6$ | $\mathbf{79.2 \pm 0.5}$ | $75.8 \pm 0.1$ | $77.2 \pm 0.3$ | $81.1 \pm 0.5$ | $78.2 \pm 0.5$ |
| | 50 | $82.3 \pm 0.3$ | $\mathbf{84.4 \pm 0.4}$ | $81.3 \pm 0.2$ | $81.8 \pm 0.2$ | $84.3 \pm 0.1$ | $82.4 \pm 0.1$ |
| CIFAR-10 | 1 | $28.3 \pm 0.5$ | $28.8 \pm 0.7$ | $59.1 \pm 0.4$ | $\mathbf{61.1 \pm 0.7}$ | $64.7 \pm 0.2$ | $61.4 \pm 0.8$ |
| | 10 | $44.9 \pm 0.5$ | $52.1 \pm 0.5$ | $67.0 \pm 0.4$ | $\mathbf{73.1 \pm 0.1}$ | $75.6 \pm 0.2$ | $73.7 \pm 0.2$ |
| | 50 | $53.9 \pm 0.5$ | $60.6 \pm 0.5$ | $71.7 \pm 0.2$ | $\mathbf{76.1 \pm 0.3}$ | $80.6 \pm 0.1$ | $76.6 \pm 0.3$ |
| CIFAR-100 | 1 | $12.8 \pm 0.3$ | $13.9 \pm 0.3$ | $31.8 \pm 0.3$ | $\mathbf{36.0 \pm 0.4}$ | $34.9 \pm 0.1$ | $44.1 \pm 0.1$ |
| | 10 | $25.2 \pm 0.3$ | $32.3 \pm 0.3$ | $\mathbf{46.0 \pm 0.2}$ | $44.9 \pm 0.2$ | $49.5 \pm 0.3$ | $46.8 \pm 0.2$ |

resulting MSE error on $x_1$ and $x_2$ would be 16 and 4, respectively. Notably, $x_1$ incurs a larger loss and results in a larger gradient on $X_S$ than $x_2$, despite being correctly classified and $x_2$ being incorrectly classified. In the heavily constrained dataset distillation setting, fitting both data points simultaneously is not possible, leading to underfitting of the data in terms of classification in order to better fit already-correctly labeled data points in terms of regression.

**Unclear probabilistic interpretation of MSE for classification.** This prevents regression from being used directly in calibration-sensitive environment, necessitating the use of transformation functions in tasks such as GP classification [Williams and Barber, 1998, Milios et al., 2018].

Based on these two issues, we adopt : Platt scaling [Platt, 2000], by applying a cross entropy loss to the labels instead of an MSE one: $\mathcal{L}_{\text{platt}} = \text{x-entropy}(y_T, f(X_T)/\tau)$, where $\tau$ is a positive learned temperature scaling parameter. Unlike typical Platt scaling, we learn $\tau$ jointly with our support set instead of post-hoc tuning on a separate validation set. $f(X_T)$ is still calculated using the same KRR formula. Accordingly, this corresponds to training a network using MSE loss, but at inference, scaling the outputs by $\tau^{-1}$ and applying a softmax to get a categorical distribution. Unlike typical GP classification, we ignore the variance of our predictions, taking only the mean instead.

The combination of these three changes, namely, using the NNGP kernel instead of NTK, applying a random-feature approximation of NNGP, and Platt-scaling result in our RFAD algorithm, which is given in algorithm 1.

## 4 Experiments with RFAD

Here, we perform experiments to evaluate the performance of RFAD in dataset distillation tasks.

**Benchmarks.** We applied our algorithm to five datasets: MNIST, FashionMNIST, SVHN, CIFAR-10 and CIFAR-100 [LeCun et al., 2010, Xiao et al., 2017, Netzer et al., 2011, Krizhevsky et al., 2009], distilling the datasets to coresets with 1, 10 or 50 images per class.

**Network Structure and Training Setup.** Similar to previous work on dataset distillation, we used standard ConvNet architectures with three convolutional layers with average pooling and ReLU activations [Zhao et al., 2021, 2020, Nguyen et al., 2021b]. Similar to KIP [Nguyen et al., 2021b], we do not use instancenorm layers because of the lack of an infinite-width analog. During training, we used $N = 8$ random models, each with $C = 256$ convolutional channels per layer, and during test-time, we evaluated the datasets using the exact NNGP kernel using the neural-tangents library [Novak et al., 2020]. We consider both the fixed and learned label configurations, with Platt scaling applied and no data augmentation. We used the regularized Zero Component Analysis (ZCA) preprocessing

Table 2: Performance of finite networks trained with gradient descent on DC/DSA, KIP, and RFAD distilled images. * denotes the result was obtained using learned labels. (n = 12)

|  | Img/Cls | DC/DSA | KIP to NN | RFAD to NN |
|---|---|---|---|---|
| MNIST | 1 | $91.7 \pm 0.5$ | $90.1 \pm 0.1$ | $\mathbf{94.4 \pm 1.5}^*$ |
|  | 10 | $97.8 \pm 0.1$ | $97.5 \pm 0.0$ | $\mathbf{98.5 \pm 0.1}^*$ |
|  | 50 | $\mathbf{99.2 \pm 0.1}$ | $98.3 \pm 0.1$ | $98.8 \pm 0.1$ |
| Fashion-MNIST | 1 | $70.6 \pm 0.6$ | $73.5 \pm 0.5^*$ | $\mathbf{78.6 \pm 1.3}^*$ |
|  | 10 | $84.6 \pm 0.3$ | $\mathbf{86.8 \pm 0.1}$ | $87.0 \pm 0.5$ |
|  | 50 | $\mathbf{88.7 \pm 0.2}$ | $88.0 \pm 0.1^*$ | $\mathbf{88.8 \pm 0.4}$ |
| SVHN | 1 | $31.2 \pm 1.4$ | $\mathbf{57.3 \pm 0.1}^*$ | $52.2 \pm 2.2^*$ |
|  | 10 | $\mathbf{79.2 \pm 0.5}$ | $75.0 \pm 0.1$ | $74.9 \pm 0.4$ |
|  | 50 | $\mathbf{84.4 \pm 0.4}$ | $80.5 \pm 0.1$ | $80.9 \pm 0.3^*$ |
| CIFAR-10 | 1 | $28.8 \pm 0.7$ | $49.9 \pm 0.2$ | $\mathbf{53.6 \pm 1.2}^*$ |
|  | 10 | $52.1 \pm 0.5$ | $62.7 \pm 0.3$ | $\mathbf{66.3 \pm 0.5}^*$ |
|  | 50 | $60.6 \pm 0.5$ | $68.6 \pm 0.2$ | $\mathbf{71.1 \pm 0.4}$ |
| CIFAR-100 | 1 | $13.9 \pm 0.3$ | $15.7 \pm 0.2$ | $\mathbf{26.3 \pm 1.1}^*$ |
|  | 10 | $32.3 \pm 0.3$ | $28.3 \pm 0.1$ | $\mathbf{33.0 \pm 0.3}^*$ |

for SVHN, CIFAR-10, and CIFAR-100, to improve KRR performance for color image datasets [Shankar et al., 2020, Nguyen et al., 2021b]. More details are available in appendix D.

**Baselines.** We compare RFAD to recently developed advanced dataset distillation algorithms such as: KIP [Nguyen et al., 2021a,b], Dataset Condensation with gradient matching (DC) [Zhao et al., 2021], and differentiable Siamese augmentation (DSA) [Zhao and Bilen, 2021b].

Table 1 summarizes the results. We observe that in the fixed label configuration, our method outperforms other models in almost every dataset. In particular, it outperforms KIP by up to $6.1\%$ in the CIFAR-10 10 img/cls setting. We attribute this gain primarily to the use of Platt scaling. RFAD falls slightly behind KIP with learned labels. While this could partially be explained because we did not apply data augmentation, which marginally elevated performance for KIP on some datasets [Nguyen et al., 2021b], we hypothesize that the performance difference is caused by the increased gradient variance associated with the random feature method. Nevertheless, in all experiments, RFAD is at least two orders of magnitude faster than KIP (Figure 2).

### 4.1 Time Savings during training

Next, we evaluated the time efficiency of RFAD. fig. 2 shows the time taken per training iteration on CIFAR-10 over coreset sizes and the number of models, $N$ used to evaluate the empirical NNGP kernel during training. Each training iteration contains 5120 examples from the training set. fig. 2 depicts that the time taken by RFAD is linear in both the number of models used during training and in the coreset size, validating the time complexity described above. We expect that for larger coreset sizes, the matrix inversion will begin to dominate due to its cubic complexity, but for small coreset sizes, the computation of the kernel matrix dominates the computation time.

In the right-hand side plot in fig. 2 we show the same plot in log-scale, compared to KIP. For KIP, we used a batch size of 5000, and rather than measuring the time taken, we use the calculation provided in appendix B of [Nguyen et al., 2021b], which describes the running time of the algorithm. We observe evidently that even for the modest coreset sizes, the quadratic time complexity of computing the exact kernel matrix in KIP results in it being multiple orders of magnitude slower than our RFAD. Both KIP and RFAD converge in between 3000-15000 training iterations, resulting in times between 1-14hrs for RFAD and several hundred GPU hours for KIP, depending on the coreset size dataset, and when the early stopping condition is triggered.

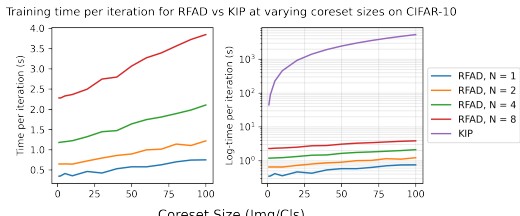

Figure 2: Time per training iteration for RFAD and KIP with varying number of models, $N$. Left: Linear plot of time. Right: Logarithmic time for training iteration. RFAD achieves over two-orders-of-magnitude speedup compared KIP per training iteration while converging with a similar number of iterations.

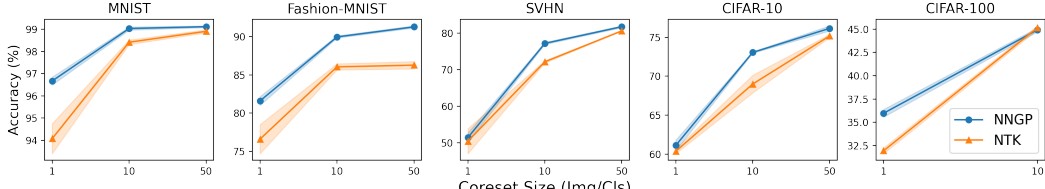

Figure 3: NNGP to NTK transfer performance on RFAD distilled images. The blue line indicates the performance of RFAD distilled images evaluated on NNGP. The orange line shows the same images evaluated using NTK. Despite being trained using the empirical NNGP kernel, these images still perform well on the NTK kernel, losing at most a few percentage points. (n = 4)

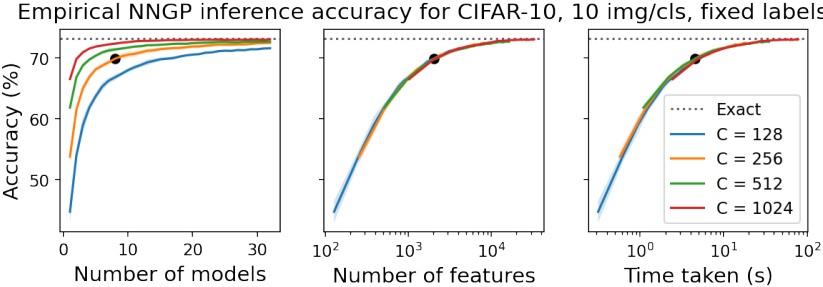

Figure 4: Empirical NNGP kernel performance at test-time with a varying number of models used to compute the empirical NNGP kernel and number of convolutional channels per model. (n = 5)

## 4.2   NTK Kernel and Finite Network Transfer

One of the key elements of the RFAD algorithm is the replacement of NTK with the empirical NNGP kernel. While we argued earlier that the two should exhibit a similar performance given their similar formalism, in this section, we verify this claim experimentally. We evaluated our distilled coresets obtained from our RFAD algorithm in two different transfer scenarios. In the first setting, at test time, we used an NTK kernel instead of the NNGP kernel. In the second setting, we trained a finite-width network with gradient descent on the distilled datasets obtained via RFAD. Similar to [Nguyen et al., 2021b], we used a 1024-width finite network for our finite-transfer results since it better mimics the infinite width setting that corresponds to the NTK.

Remarkably, as shown in fig. 3, in most datasets, these coresets suffer little to no performance drop when evaluated using NTK compared to the exact NNGP kernel, despite being trained using the empirical NNGP kernel. The largest performance gap is 8% on SVHN with 10 images per class, and in some datasets, notably CIFAR-100, 10 img/cls evaluating using the NTK kernel outperforms NNGP. This suggests that either the exact NNGP kernel or the random feature NNGP kernel could potentially be used as a cheaper approximation for the exact NTK kernel.

table 2 shows the resulting finite network transfer when training with gradient descent on our coresets. Our images appear to have the best performance in finite-network transfer, outperforming KIP in almost all benchmarks and the DC/DSA algorithms in many, despite DC/DCA being designed specifically for finite-width networks. We attribute this performance gain over KIP primarily to two tricks we used during training. Firstly, we applied centering, which, rather than training a typical network $f_\theta(x)$, we instead train a network with its output at initialization subtracted: $f_\theta(x) - f_{\theta_0}(x)$.

This has been shown empirically to speed up the convergence of finite-width networks by reducing the bias caused by the finite-width initialization while still preserving the NTK [Lee et al., 2020, Hu et al., 2020]. We find that for these small datasets, this modification significantly improves performance. The second trick is label scaling; we scale the target labels by a factor $\alpha > 1$: $\mathcal{L}_\alpha = ||f_\theta(x) - \alpha y||_2^2/\alpha^2$, and at inference divide the model's outputs by $\alpha$. Note that this does not

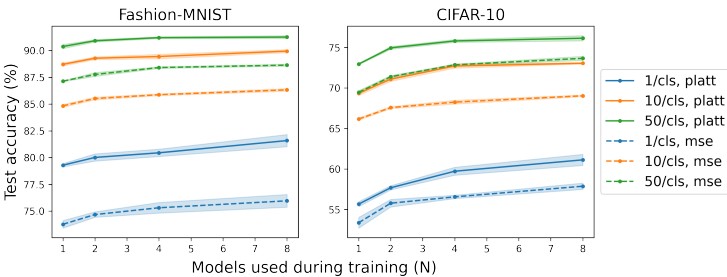

Figure 5: Objective function sensitivity. (n = 4) to use Platt scaling and the number of models during training (N) for Fashion-MNIST and CIFAR-10

affect the infinite-width setting, as in KRR, the output is linear w.r.t. the support set labels. Ablations of these changes are in appendix F.

### 4.3 Empirical NNGP Performance at Inference

To validate the efficacy of our method, we evaluated our distilled coresets using features from random networks as opposed to the exact kernel. We varied the width of individual networks between 128 and 1024 channels and the number of models between 1 and 32. fig. 4 shows the resulting classification accuracy on the CIFAR-10 dataset with 10 images/class. The black dot represents the configuration we used during training: 8 models, each with width 256 (More ablations are provided in appendix I). We conclude that the random feature method, for all network widths, is able to reach close to the exact NNGP kernel performance (dotted line) if a sufficient number of models are used. Interestingly, the performance is almost entirely dependent on the total number of features (proportional to $C \times N$, with $C$ being the number of convolutional channels) and not the width of individual networks, suggesting that the finite-width bias associated with random finite networks is minimal.

In appendix I, we show that this can be taken to the extreme, with $70\%$ accuracy achieved with a network with a *single* convolutional channel. These results corroborate the findings of [Novak et al., 2019], which first proposed this random feature method, where they found, like us, that performance was almost entirely determined by the total feature count.

Platt scaling. We performed ablations on the use of the cross-entropy loss and the number of models used during training. We reran our algorithm on CIFAR-10 and Fashion-MNIST, using either 1, 2, 4, or 8 models during training, using MSE loss or cross-entropy loss. fig. 5 shows the resulting performance of these configurations. Evidently, using a cross-entropy loss results in substantial performance gains, even as much as $8\%$ as with Fashion-MNIST with one img/cls.

## 5 RFAD Application I: Interpretability

Large datasets contribute to the difficulty of understanding deep learning models. In this paper, we consider interpretability in the sense of the influence of individual training examples on network predictions [Hasani et al., 2019, Lechner et al., 2020, Wang et al., 2022]. One method of understanding this effect is the use of influence functions, which seek to answer the following counterfactual question: which item in the training set, if left out, would change the model's prediction the most [Hampel, 1974, Koh and Liang, 2017, Kabra et al., 2015]? For deep networks, this can only be answered approximately. This is because retraining a network on copies of the training set with individual items left out is computationally intractable. One solution is to use kernel ridge regression on a small support set. We can recompute the KRR on the kernel matrices with the $i$th individual coreset element removed, with $K_{x,S\setminus i}$ $K_{S\setminus i,S\setminus i}$ being the resulting kernel matrices with the $i$th row/column corresponding to the $i$th coreset entry removed.

In particular, let $p(y_{\text{test}} = c|S)$ be the probability prediction (computed by applying Platt scaling) of an example belonging to class $c$ computed on the entire coreset, $S$. Let $p(y_{\text{test}} = c|S \setminus i)$ be the same prediction calculated with the $i$th coreset element removed. We define the influence score, $I_i$ of coreset element $i$ on $x_{\text{test}}$ as $\sum_{c \leq C} |p(y_{\text{test}} = c|S) - p(y_{\text{test}} = c|S \setminus i)|$. Taking the top values of $I_i$ yields the most relevant examples.

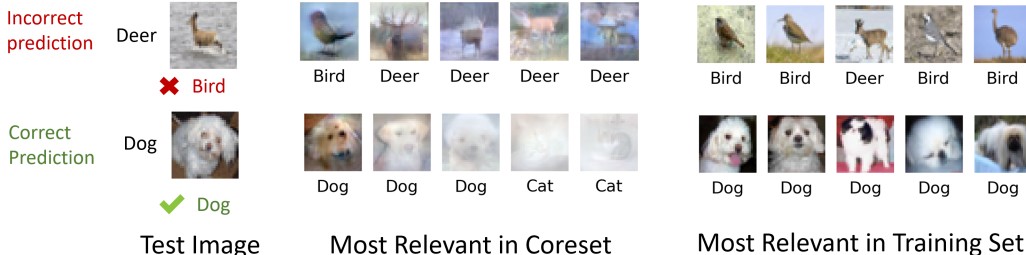

| Incorrect prediction | Deer | Most Relevant in Coreset | | | | | Most Relevant in Training Set | | | | |
|---|---|---|---|---|---|---|---|---|---|---|---|
| | ❌ Bird | Bird | Deer | Deer | Deer | Deer | Bird | Bird | Deer | Bird | Bird |
| Correct Prediction | Dog | | | | | | | | | | |
| | ✅ Dog | Dog | Dog | Dog | Cat | Cat | Dog | Dog | Dog | Dog | Dog |
| | Test Image | Most Relevant in Coreset | | | | | Most Relevant in Training Set | | | | |

Figure 6: The most relevant images for an incorrect (top row) and correct (bottom row) prediction on CIFAR-10. The most relevant coreset images are picked based on the coreset influence score $I$, and for their training set, the training influence score $J$. These metrics are fast to compute and result in semantically meaningful explanations for these two predictions.

While this method provides a simple way of gaining insights into how a prediction depends on the coreset, it does not provide insight into how this prediction comes from the original training set which produced the coreset. The method can be extended to accommodate this. Heuristically, we conjecture that two elements are similar if their predictions depend on the same elements in the coreset. We compute $p(y_j = c|S)$ and $p(y_j = c|S \setminus i)$ for every element $j$ in the training set and $i$ in the coreset. Then, we define its influence embedding as $z_{i,c}^j = p(y_j = c|S) - p(y_j = c|S \setminus i), z^j \in R^{|S| \times |C|}$. This way, $z^j$ defines the sensitivity of a training datapoint prediction on the coreset. We compute the same embedding for a test datapoint $z^{\text{test}}$, and to compare data points we use cosine similarity, $J_{\text{test},j} = \cos(z^{\text{test}}, z^j)$. Values of $z^j$ can be precomputed for the training set, typically in a few minutes for CIFAR-10, allowing for relatively fast queries, in contrast to the more expensive Hessian-inverse vector product required in Koh and Liang [2017], which is costly to compute and difficult to store.

fig. 6 shows the results of this algorithm applied to CIFAR-10 with 50 img/cls for an incorrectly and correctly predicted image. In both cases, the resulting queries are visually similar to the test data point. One could use this information to not only explain a single incorrect prediction but to understand harmful items in their test set, or where more data needs to be collected.

As a second application of RFAD, we create coresets that contain no human-understandable information from the source dataset yet still retain high test performance. [Nguyen et al., 2021a] proposed the concept of a $\rho$-corrupted coreset: A fraction $\rho$ of the coresets elements are completely independent of the source dataset. Practically, for our algorithm, this means initializing the coreset with random noise and keeping a random $\rho$ fraction of the pixels kept at their initialization. We term this algorithm RFAD$_\rho$.

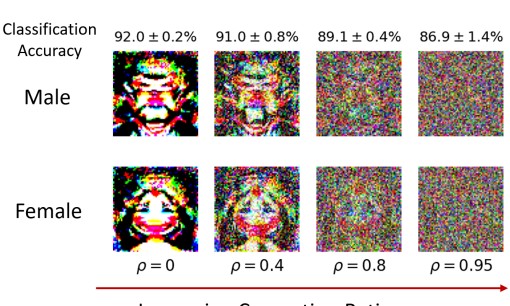

| Classification Accuracy | $92.0 \pm 0.2\%$ | $91.0 \pm 0.8\%$ | $89.1 \pm 0.4\%$ | $86.9 \pm 1.4\%$ |
|---|---|---|---|---|
| Male | | | | |
| Female | | | | |
| | $\rho = 0$ | $\rho = 0.4$ | $\rho = 0.8$ | $\rho = 0.95$ |

Increasing Corruption Ratio, $\rho$

Figure 7: CelebA distilled datasets for male/female classification with 1 image per class at varying corruption ratios. At $\rho = 0$, the distilled images are very interpretable, but at $\rho = 0.95$, the images look like white noise, despite achieving 86.9% accuracy on the classification task.

Adding noise to gradient updates of the inputs of a network can be shown to give differentially private guarantees [Abadi et al., 2016]. While our scheme does not provide the same guarantees, we note the following two privacy-preserving properties of RFAD$_\rho$: firstly, the distillation process is irreversible: there are many datasets for which a distilled dataset provides zero loss. Secondly, if the true data distribution assigns a low probability to images of white noise, then for high values of $\rho$, this guarantees that the distilled dataset stays far away in $L_2$ norm from real data points, since $\rho$ fraction of the pixels are stuck at their initialization. This means that a distilled RFAD$_\rho$ dataset will not recreate any real points in the training set.

# 6 RFAD Application II: Privacy

We applied RFAD$_\rho$ on CIFAR-10, and CelebA faces datasets. For CIFAR-10 we distilled the standard 10-class classification task, with corruption ratios taking values of $[0, 0.2, 0.4, 0.8, 0.9]$, with 1, 10 or 50 images per class. For CelebA, we performed male/female binary classification with corruption ratios between 0 and 0.95 with 1, 10, or 50 samples per class. fig. 8 show the resulting performance.

For CIFAR-10, even at corruption ratios of 0.9, we are able to achieve 40.6% accuracy with one sample per class, far above the natural baseline of 16.1% ([Nguyen et al., 2021b] table A1). For CelebA, we achieve 81% accuracy with only two images, one male and one female, with 95% of the pixels in the image being random noise.

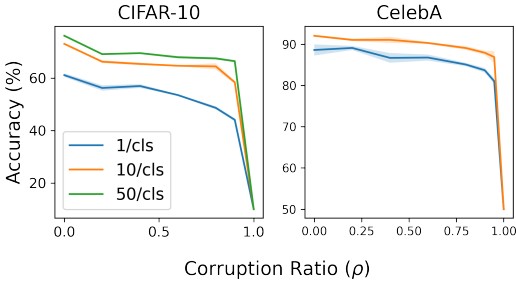

Figure 8: Performance of RFAD on CIFAR-10 and CelebA classification with varying support set sizes and corruption ratios. Performance degrades very gradually as noise is increased, still achieving high performance with 90% corruption. (n = 3)

We additionally visualize the distilled images for the male and female classes in the CelebA distillation task with one image per class in fig. 7 at varying corruption ratios. While the image initially contains visually interpretable features with $\rho = 0$, they quickly devolve into pure noise at $\rho = 0.95$.

# 7 Conclusions

We proposed RFAD, a dataset distillation algorithm that provides a 100-fold speedup over the current state-of-the-art KIP while retaining accuracy. The speedup is primarily due to the use of the approximate NNGP kernel as opposed to the exact NTK one, reducing the time complexity from $O(|S|^2)$ to $O(|S|)$. The success of the approximation here, combined with the similarity between the NTK and NNGP kernels, suggests the random network NNGP approximation as an efficient method for algorithms where the exact computation of the NNGP or NTK kernel is infeasible. We analyzed our method comprehensively and showed its effectiveness, and proposed two applications in model interpretability and privacy preservation. With this new tool, we hope that future work could begin to use Neural Tangent Kernel as an algorithmic design tool in addition to its current theoretical use for neural network analysis. Lastly, RFAD has the following limitations:

**Use of instancenorm.** In practice, we found that our datasets distilled without instancenorm do not transfer well to finite networks with it. Conversely, if we use random networks with instancenorm in RFAD, these transfer to finite networks with instancenorm but not to ones without or the NNGP kernel. This suggests that the features used by networks with/without instancenorm differ, making it difficult to distill datasets that perform well on both. We discuss this further in appendix G.

**Overfitting in dataset distillation.** On Platt scaling, we argued that the heavily constrained nature of dataset distillation leads to underfitting of the training set when using an MSE-style loss in KIP, and we verified the efficacy of using a Platt loss instead. However, we observed that in simple datasets, such as MNIST, or with large coresets relative to the data, such as CIFAR-100 with 10 images per class, we could overfit to the dataset. We found that these distilled datasets were able to achieve near 100% classification accuracy on the training set, meaning that it was distilled perfectly in terms of the Platt-loss. This implies that adding more images would not improve performance. Thus we hypothesize that using a Platt loss would be detrimental if the compression ratio is low.

# Acknowledgements

This research has been funded in part by the Office of Naval Research Grant Number Grant N00014-18-1-2830, DSTA Singapore, and the J. P. Morgan AI Research program. We are very grateful.

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
