# OpenReview forum: "Efficient Dataset Distillation using Random Feature Approximation"
_NeurIPS.cc/2022/Conference — NeurIPS 2022 Accept_

### Official Review · Reviewer_DFgr · 2022-07-08

**Rating:** 7
**Confidence:** 4
**Soundness:** 4 excellent
**Presentation:** 3 good
**Contribution:** 3 good

**Summary:**

The paper provides a data distillation inspired by KIP. While KIP is the current state-of-the-art data distillation model, the complexity of such an algorithm becomes impractical when dealing with large models as well as large datasets. The authors refer to the use of an approximated variant of the NNGPs via Random Features to accelerate the KIP algorithm while maintaining comparable results.

**Questions:**

1) Using a different loss than the MSE loss function is indeed clever. However, would it be better to use a more robust loss function than that of the MSE, for example, $\ell_1$-loss for example?
2) Is it possible to run your algorithm with coreset_size=50 with CIFAR-100? If not, is it related to insufficient memory?
3) is the proposed method applicable when using the Imagenet dataset or a more complex deep learning model, e.g., ResNet50?


**Limitations:**

The authors clearly stated their limitations.

**Strengths And Weaknesses:**

* Strengths:
    1) The paper presents a practical algorithm for dataset distillation.
    2) It was a great idea to introduce an alternative loss function, replacing the MSE to obtain better results.
    3) The authors shed light on the applicabilities of their methods, from privacy, interpretability, and data corruption.
    4) The paper is clear and properly written.

* Weaknesses:
    1) From a theoretical point of view, what is the approximation factor of the approximated NNGP kernel matrix with respect to the NNGP kernel matrix? Please do shed some light on this.
    2) Similar to KIP, only coresets of size 10 were computed when handling the CIFAR-100 dataset. See Questions.

---

> ### Author Response · Authors · 2022-08-02
> **Reviewer DFgr response**
>
> We would like to thank the reviewer very much for their positive evaluation of our work and for their thoughtful review. Here we address the concerns and questions raised.
>
> **Approximate NNGP error:** We discuss the asymptotic bias in lines 132-138. The bias of the approximation is O(1/C), with C being the network width. In practice we found this bias to be small, and we can achieve performance similar to the exact NNGP kernel with a modest number of channels (256), as we show in section 4.3 and appendix I. The exact constant of this bias is difficult to calculate in closed form, and can be found in [4].
>
> **CIFAR-100 coreset size:** Indeed coresets of size 50 for CIFAR-100 exceed the memory budget for a single GPU (24GB in our case). A limitation of our algorithm is that it requires calculating the kernel matrix for the entire coreset in each forward pass, meaning that it has O(|S|) memory. Note however that in comparison to KIP, hundreds of GPUs are required per iteration for even CIFAR-10 coreset sizes 10 (See [1] Appendix B), so there is already a dramatic memory savings despite this limitation. It is likely a batching approach would allow the algorithm to scale to larger coresets, as it has been applied to other algorithms which calculate the NTK [3], but we leave this to future work.
>
> **Different loss functions:** We tried a few different loss functions, such as L1, hinge loss, etc and found that cross-entropy worked the best. L1 performed similarly to the standard MSE loss.
>
> **Larger datasets/models:** Yes, our algorithm should in principle work with larger datasets/models, however it is worth noting that a key limitation of NTK-based algorithms for deeper models becomes the weaker correspondence between the NTK and practical sized networks [2]. Future work could look at closing this gap.
>
> [1] Nguyen, Timothy, et al. Dataset Distillation with Infinitely Wide Convolutional Networks. arXiv:2107.13034, arXiv, 17 Jan. 2022. arXiv.org, http://arxiv.org/abs/2107.13034.
>
> [2] Lee, Jaehoon, et al. “Finite versus Infinite Neural Networks: An Empirical Study.” Proceedings of the 34th International Conference on Neural Information Processing Systems, Curran Associates Inc., 2020, pp. 15156–72.
>
> [3] Yuan, Chia-Hung, and Shan-Hung Wu. “Neural Tangent Generalization Attacks.” Proceedings of the 38th International Conference on Machine Learning, edited by Marina Meila and Tong Zhang, vol. 139, PMLR, 2021, pp. 12230–40, https://proceedings.mlr.press/v139/yuan21b.html.
>
> [4] Roberts, Daniel A., et al. The Principles of Deep Learning Theory. June 2021. arxiv.org, https://doi.org/10.1017/9781009023405.

---

### Official Review · Reviewer_ta1Y · 2022-07-10

**Rating:** 6
**Confidence:** 4
**Soundness:** 4 excellent
**Presentation:** 3 good
**Contribution:** 3 good

**Summary:**

The paper proposed an approximation algorithm to accelerate the Kernel Inducing Points (KIP) method with neural kernels for dataset distillation problem, which is mainly based on the random feature approximation of the Neural Network Gaussian Process (NNGP) kernel. To save the time cost of the KIP algorithm with neural kernels, the paper focused on approximating the NNGP kernel instead of the Neural Tangent Kernel (NTK), and proposed a simple approximation algorithm with empirical NNGP features which are easily generated from finite-width random neural networks. Besides, the paper changed the square loss to cross-entropy loss when learning the inducing points. In experiments, the efficiency superiority of the proposed method RFAD is verified with limited loss on performance. In addition, despite training KIP on the NNGP kernel which is naturally friendly for computation, the obtained distilled data could also apply to NTK and finite-width neural networks.

**Questions:**

1. What does it mean by "n=4"|"n=12" in the tables?
2. What is the difference between fixed and learned label configurations? For fixed label configuration, do you mean the label solving (LS) method?
3. As claimed in the experiments section, the performance gain is primarily attributed to the use of Platt scaling. Does this mean that if using cross-entropy loss, original KIP will perform even better than the results in tables?
4. It looks that replacing the loss function is independent to the speeding up of the algorithm, thus a comparison with RFAD without cross-entropy loss, or KIP with cross-entropy loss is necessary.
5. What are the technical challenges in the implementation? They should be written in the paper.

minor issues:

Page 8 and Page 4. The subscripts of the letters in the formulas are too large.

Besides, compared to many related work about coresets, Reduced Set Construction is more similar to this paper and dataset distillation, which was studied in:

Scholkopf, B.; Mika, S.; Burges, C. J. C.; Knirsch, P.; Muller, K.; Ratsch, G.; and Smola, A. J. 1999. Input space versus feature space in kernel-based methods. IEEE Transactions on Neural Networks 10(5): 1000–1017.

**Limitations:**

NA.

**Strengths And Weaknesses:**

Stengths:

1. The efficiency superiority of the proposed method RFAD is impressing. The reduction on the computational cost for KIP on NNGP over MNIST and CIFAR is significant compared to prior works [Nguyen et al., 2021], and succeed to make the KIP with neural kernels more practical for real scenarios. The engineering efforts should be admired.

2. The paper is technically sound and well presented with broad literature review. Extensive experiments could support the effectiveness of the proposed method.

Weakness:

1. The main weakness of the paper is: the novelty is limited, especially compared with the previous work [Nguyen et al., NeurIPS'21; Nguyen et al., ICLR'21]. The approximation scheme is direct, and theoretically guaranteed, and already verified in the experiments of existing work. Besides, the improvement on MNIST and CIFAR is significant in [Nguyen et al., 2021], and before that we can not imagine that one can get 64.3% performance on CIFAR with only 1 image per class. However, from training KIP with KRR on NTK, to training KIP with KRR on NNGP, and then to training KIP with KRR on random finite-width neural networks and cross-entropy loss. The theoretical rigor is getting weaker and weaker.

2. The Platt scaling attributed highly to the performance of RFAD, but not compared separately.

---

> ### Author Response · Authors · 2022-08-02
> **Reviewer ta1Y response**
>
> We would like to thank the reviewer for their review of our work.
> In the following we will provide a detailed response to the raised concerns and hope to motivate a more positive assessment of our work, aligned with other reviewers’ positive evaluation.
>
> **Limited novelty:** We believe that while the findings and technical modifications applied to the kernel-induced points algorithm for dataset distillation are straightforward, it is exactly this simplicity which signifies the effectiveness of our approach to make dataset distillation algorithms accessible and tractable to a broad scheme of applications weren’t possible before.
>
> We consider our work as an important algorithmic contribution to enable future research on dataset distillation at scale. We hope that the reviewer agrees with us and other reviewers.
>
> Nevertheless, and in particular, the use of random feature approximation as opposed to exact kernel calculation is novel and significant especially as it shows orders of magnitude benefit in obtaining distilled datasets without loss of accuracy. Moreover, in regard to novelty and significance, we would like to add that:
>
> 1. Besides the technical novelty, our method reduces the need for compute from over 100 GPUs for dataset distillation, to a single GPU.
>
> 2. This is a direct result of our algorithmic contribution that speeds up kernel-induced points by over two orders of magnitude.
>
> 3. Our algorithm achieves this speed-up without performance loss.
>
> As the reviewer themselves pointed out: “The efficiency superiority of the proposed method RFAD is impressing.”,  we believe that our results are not only impressive, but turning an otherwise unscalable and non-tractable fundamental algorithm (such as KIP) into one that can scale and be computationally feasible, with simple and elegant modifications, should justify enough novelty from both technical and engineering standpoints. Therefore, we would like to ask the reviewer to consider the simplicity of our proposed method as a strong advantage rather than a setback.
>
> **Platt scaling evaluation:** We showed the relative effect of platt scaling in Figure 6 and in line 271. Indeed, platt scaling does result in a significant performance gain for our algorithm (several percent). It is possible that KIP would benefit from this change as well, however **the high computational cost of KIP prevents us from testing this**. For example, for CIFAR-10 with 10 images/class KIP takes 460 GPU-seconds per interaction, compared to our 2.4s. Quoting from [1] appendix B, KIP has to use “hundreds of GPUs working in parallel, [which] enables training to become computationally feasible.”
>
> For RFAD, we hypothesize that platt scaling helps reduce some of the gradient variance associated with the random feature approximation which does not appear in KIP. This hypothesis has to be tested which we will do in our continued effort. Accordingly, in theory, we believe that KIP would see a lesser improvement from using platt scaling.
>
> With MSE, the gradients scale with the error, meaning that for large errors the gradients can change dramatically, whereas with cross-entropy the gradients saturate as the error gets large. If we have a noisy and large error for certain samples, MSE loss will suffer from greater gradient variance than cross-entropy.
>
> We will certainly elaborate further on this matter and include this technical discussion in our revised version.
>
>
> **Questions**
>
> 1. “N = __” in tables: This refers to the number of independent evaluations used to generate means and variances in the tables.
> This is not exactly the same as label solve as LS finds the minimum label value exactly and does not optimize the images themselves. In our setting we jointly optimize the image pixels and labels values via gradient descent
>
> 2. KIP is computationally intractable therefore we did not test KIP with Platt Scaling. See Platt scaling evaluation comment
>
> 3. See Platt scaling evaluation comment
>
> 4. Technical challenges: Appendix D discusses our implementation details. The implementation of RFAD algorithm is relatively straightforward, making it an appealing algorithm for dataset distillation across different applications. We also denote that using double precision in the matrix inversion is necessary for obtaining the best performance.
> We will add this note to our revised version.
>
> We will also add the citation to Scholkopf, et al 1999, in our revised version.

---

> > ### Comment · Reviewer_ta1Y · 2022-08-07
> > **Thanks for the response.**
> >
> > I have read all the reviews and rebuttal. The authors did a good job in answering my concerns about novelty with detailed impressing results. Although the main contribution is not primarily technical, I believe that I underestimated the contribution of the paper (engineering), especially making the KIP scheme tractable without loss of accuracy. I will accordingly revise my score.
> >
> > Please allow me several follow-up questions.
> >
> > Although the random fourier features (RFF) have good theoretical approximation properties, in experiments, we have to compromise the performance when using approximation. Besides, the Nystrom-based method usually performed even better than RFF method [1]. But in both schemes we have to lose performance. Why could you accelerate so much in speed without losing accuracy? Could you provide some insight?
> >
> > Moreover, there are some existing studies for scaling NTKs like [2], what are the difference between them and this paper? I believe this should also be discussed in the paper.
> >
> > [1] Tianbao Yang, et al. "Nyström method vs random fourier features: A theoretical and empirical comparison." NeurIPS 2012.
> >
> > [2] Amir Zandieh, et al. "Scaling neural tangent kernels via sketching and random features." NeurIPS 2021.

---

> > > ### Author Response · Authors · 2022-08-09
> > > **Reviewer Response**
> > >
> > > We would like to thank the reviewer very much for acknowledging our main contribution, updating their score, and voting for the acceptance of our submission.
> > >
> > > In the following we provide answers to their follow-up questions:
> > >
> > > **Why is there is little to no drop in performance:** Unfortunately we do not have an exact explanation as to why the drop in performance from using the empirical NNGP is so small. We hypothesize that during optimization, small amounts of gradient variance do not affect optimization significantly, as evident with the success of stochastic gradient methods for optimizing neural networks. Unfortunately, in our case we are unable to show that the gradients associated with the empirical NNGP are unbiased, however we found that in practice the bias is very small. As we see with section 4.3, figure 5, and the appendix figures 13-22, we can achieve high performance with the empirical NNGP approximation with sufficient total numbers of features (irrespective of the width of individual models). Future work could look at quantifying the exact bias and variance associated with these gradients estimates, especially in the case where optimizing the empirical NNGP becomes a common algorithmic tool.
> > >
> > > **Comparison with other NTK scaling schemes:**  Indeed [1] shows impressive results for scaling the NTK computation, however their approach may not be as useful for dataset distillation. Firstly, note the time taken in table 1 in [1]. Their method (CNTKSKETCH) is at least twice as slow as GradRF (gradient random features). Our method, using the empirical NNGP, is significantly faster than GradRF, as it does not require calculating parameter gradients, and only requires forward passes through the network. While their method would likely produce higher accuracy during inference time than ours, as we have shown throughout our paper, during training, lower accuracies do not significantly affect the final coreset performance. Second, their method is significantly more complex to implement, and requires careful tuning and re-engineering for different architectures, layers types, and activation functions, while our method is model-agnostic. This lets us quickly change architectures and model parameters without worrying about re-deriving new sketching rules.
> > >
> > > [1] Amir Zandieh, et al. "Scaling neural tangent kernels via sketching and random features." NeurIPS 2021.

---

### Official Review · Reviewer_orEs · 2022-07-12

**Rating:** 6
**Confidence:** 3
**Soundness:** 3 good
**Presentation:** 3 good
**Contribution:** 3 good

**Summary:**

The paper proposes a set of changes/improvements to a prior SOTA dataset distillation algorithm (KIP), that allows to achieve comparable or better results at a fraction (e.g. 1/100) of the compute cost.

The changes are:
- Using finite-width NNGP at training time (vs the infinite-width NTK), allowing a dramatic compute improvement.
- Using the infinite-width NNGP at test time (vs the infinite-width NTK).
- Using cross-entropy loss with platt-scaling and a trainable temperature parameter.
- Representing the optimized support set as a matrix product (trainable preprocessing) instead of a single matrix.
- Using centering and label scaling for RFAD -> NN transfer.


**Questions:**

- In Table 2, are random seeds used to initialize RFAD, and those used to initialize NNs, completely disjoint? To make sure there is no benefit gained due to seed reuse.

- Line 619 "this does not add any extra variables to the coreset" - why not, doesn't it add new $(CHW)^2$ trainable parameters?

- Why do you present results with centering and label scaling in Table 2? Wouldn't it be more fair to not use these settings, since I assume they weren't used in KIP or DC/DSA (and/or rerun those methods with centering and label scaling), to which you compare? This way we could better disentangle the benefits of the images themselves vs these optimization settings.


**Limitations:**

This work will help lower the carbon footprint of the industry, and make dataset distillation algorithms more accessible to the broader community, so I believe it only has a positive societal impact.


**Strengths And Weaknesses:**

The paper clearly demonstrates a dramatic speedup in achieving comparable to prior SOTA results, which is why I recommend acceptance. However, it does so with a moderately-sized bag of various tricks, and I find the analysis/interpretation/ablations of the results somewhat lacking. Therefore my recommendation is only weak.

## Strengths:

- Clearly demonstrated comparable or better to SOTA results at a fraction of the compute cost.
- The paper overall is clear and easy to follow.
- Code is released.

## Weaknesses:

- The compute cost analysis ($|S|$ vs $|S|^2$; section 4.1, abstract, conclusion, etc) appears vague and/or potentially wrong.
	* Firstly, the asymptotic inference costs of both KIP and RFAD are the same due to using the infinite-width kernel; RFAD could be a constant 2x faster than KIP due to using NNGP vs NTK. For a test set of size $T$, this inference cost would be something like $|S|^3 + (|S|^2 + |S||T|)H^2 W^2 D$, i.e. the cost of the support set matrix inversion plus the cost of computing the support/test set kernels.
	* At training, cost of a single KIP iteration should be same as above (where $T$ is the training set size; I think complexity on line 105 is incorrect), while RFAD would be something like $|S|^3 + (|S|^2 + |S||T|) N M + (|S|+|T|)N M^2D$, if $M$ is the number of channels (i.e. the cost of support set matrix inversion plus the cost of computing the NNGP kernels, plus the cost of computing the features for the NNGP kernel). The source of the speedup might be the relatively small $N M$ compared to $H^2 W^2 D$; but IIUC for large $|T|$, both methods remain linear in $|S|$. So I don't see immediately why KIP has different scaling with $|S|$ that your method, and how exactly you plot the KIP curve in Figure 2.
    * My quick and rough math above might be wrong, but either way the authors should provide a more rigorous complexity analysis of both methods, and revise or substantiate their claims.

- No ablations of RFAD with and without Platt scaling cross-entropy loss, and with/without trainable matrix product (as described in the appendix).

## Minor:

- Some minor typos, e.g. Line 691: "Empirical NTK" -> "Empirical NNGP". Line 676 - capitalize "Table 5". Line 671 - "our times", line 149: "Probabilistic interpretation of MSE for classification?" etc.

- Could be useful to have an ablation table with results obtained from doing inference on the test set with the same exact NNGP as used during training (I assume you have these results from the appendix figures).

- Would be interesting to also compare against the DC/DSA compute costs.

- Codebase uses two frameworks (JAX and PyTorch), which is slightly inconvenient.

---

> ### Author Response · Authors · 2022-08-02
> **Reviewer orEs response**
>
> We would like to thank the reviewer very much for their positive evaluation of our work and properly acknowledging our core contributions. Here we address their main concerns and hope to motivate acceptance of our manuscript:
>
> **Runtime analysis:** The analysis provided by the reviewer is correct but misses a few key details which we will explain and clarify in our revised manuscript:
>
> Firstly, the first two terms in the decomposition you provided (O(|S|^3) matrix inversion and O(|S|^2 + |S||T|) matrix computation are very negligible compared to the third term, feature computation, for our algorithm. Note that the matrix computation term for our algorithm simply corresponds to multiplying two S x NM and NM x T matrices together, which is fast. This means that term 3 dominates the complexity, giving it O(|S| + |T|) complexity
>
> Secondly, please denote that KIP does not have the third term in your decomposition, but in contrast the middle term has the largest coefficient, resulting in O(|S|^2 + |S||T|) complexity. This is the cause of the speed discrepancy in Figure 2, both in terms of linear and logarithmic time.
>
> Third, in practice, at each training we don’t use the full training set but instead take a batch of samples |B|, which is typically on the order of 10^3 (for both KIP and RFAD). Furthermore, for RFAD, we can perform faster computation of the training set samples as we do not need to retain the network graph for gradient computation (as the features of the training set are constant w.r.t. the coreset). This in practice makes the memory requirements of computing the training batch random features very small relative to the coreset random features. Note that because KIP does not use a feature decomposition of the kernel matrix, they cannot save memory in this way.
>
> **Platt scaling ablations:** Please see line 271 and figure 6. Admittedly the figure 6 caption is rather scant due to limited space. We will certainly elaborate further in our revised version.
>
> **Number of trainable parameters:** Yes this does increase the number of trainable parameters, but the coreset size remains the same. This is similar to parameterizing a vector as Ax, where A and x are learnable vs x. When the coreset is used, the trainable matrix is simply multiplied into the coreset values and discarded, the trainable matrix is not used for inference. In practice we found that this matrix only speeded up initial convergence but long-run performance was similar.
>
> **Label scaling in table 2:** The comparison is already muddied as KIP uses a mix of MSE and cross-entropy, DC/DSA uses cross-entropy and instancenorm and data augmentation. Because of these difficulties in comparison, we decided to compare the best case performance of all the algorithms. We have ablations for these in Appendix F.
>
> We hope to have addressed the remaining concerns of the reviewer. We are more than happy to provide further clarifications if needed.
>
> **Minor comments**:
> Typos - We will correct these for the camera-ready revision. Thank you for pointing them out
>
> Inference on exact NNGP used during training - See figure 5 in the main text and figures 13-22 in the supplementary material. The black dots correspond to the same approximate NNGP configuration used in the training. Note that we initialize new random neural networks for every iteration in training.
>
> DC/DSA compute costs - See figure 4 of [1]. We see that DSA takes ~15 hours, which is slightly slower than our algorithm, while ours obtains higher performance
>
> Two different frameworks - We will consider releasing a native JAX version in the future
>
>
> **Questions**:
> 1. The random seeds are completely disjoint.
> 2. See “number of trainable parameters” point
> 3. See label scaling point
>
> [1] Zhao, Bo, and Hakan Bilen. Dataset Condensation with Distribution Matching. arXiv:2110.04181, arXiv, 21 Apr. 2022. arXiv.org, http://arxiv.org/abs/2110.04181.

---

> > ### Comment · Reviewer_orEs · 2022-08-08
> > **Satisfied overall, but still confused with complexity analysis**
> >
> > Thank you for your replies!
> >
> > I am overall satisfied, and continue to recommend acceptance.
> >
> > However, I still find the complexity analysis misleading. Even when computing complexity of a single step (which, by the way, is a somewhat distracting metric, since we only care about the total runtime) with batch size $|T| \sim 10^3$, we have $|T| \geq |S|$, and therefore $|S|^2 + |S| |T| \sim |S| |T|$, (since in the settings that you consider, $|S|  \leq 10^3$). Therefore both methods are linear in $|S|$ under your assumptions, but you do get the speed-up since, I assume, $|S| |T| H^2 W^2 D > |T| N M^2 D$. From my understanding, your $|S|$ vs $|S|^2$ argument would emerge if $|S| \sim |T|$, but in the dataset distillation application we want specifically $|S| \ll |T|$. Therefore I still expect the camera-ready version to significantly expand and formalize the complexity analysis, and to change the $|S|$ vs $|S|^2$ framing (unless I am still misunderstanding something).

---

> > > ### Author Response · Authors · 2022-08-09
> > > **Reviewer response**
> > >
> > > Thank you very much for engaging with us during the discussion period and for your constructive follow-up questions and valuable comments.
> > >
> > > **Time Complexity** In the following, we will use |B|, the training batch size, in this case as opposed to |T|, the training set size, to avoid conflating the two. Yes, typically |B| ~ 10^3, which is usually larger than our distilled dataset sizes, but note that for CIFAR-100 with 10 samples per class, |S| approaches similar values to |B|. To be as clear as possible, we will provide runtime analysis of KIP and RFAD for the |B| > |S| and |B| < |S| regimes. In all cases we assume that the matrix inversion cost is small relative to matrix/feature computation.
> > >
> > > **Case 1: RFAD with |B| < |S|** $O(|S|NMD)$ coreset random feature computation dominates, while the $K_{SS}$ matrix computation (multiplying feature vectors together) is cheap. The resulting complexity is effectively $O(S)$
> > >
> > > **Case 2: KIP with |B| < |S|** $O(S^2)$ computation of $K_{SS}$ is the dominant term, in which case the resulting time complexity is effectively $O(S^2)$. We believe the reviewer already agrees with this statement.
> > >
> > > **Case 3:  RFAD with |B| > |S|** $O(|S|NMD)$ and $O(|B|NMD)$ random feature computation for the coreset and training batch are both are relatively expensive. Again, multiplying the random feature matrices together is cheap. The resulting complexity is thus $O(\lambda_1 |B| + \lambda_2 |S|)$ As mentioned in the previous response, training batch random features can be computed more quickly than coreset random features as the computation graph does not need to be retained for backpropagation, so $\lambda_2 >> \lambda_1$.
> > >
> > > **Case 4: KIP with |B| > |S|** $O(|B||S|)$ computation of $K_{BS}$ is the dominant term, in which case the resulting time complexity is $O(|B||S|)$, meaning linear complexity, as the reviewer pointed out. **However, since $|B| > |S|$, this is effectively worse than $O(|S|^2)$ scaling.** Therefore, KIP is only linear in a regime where linear scaling is **worse** than quadratic scaling, due to a high time coefficient associated with the linear term. Note that unlike RFAD, we cannot compute $K_{BS}$ entries any quicker than $K_{SS}$ entries, as we cannot factorize it into coreset and training batch features, meaning we must retain the entire computation graph.
> > >
> > > **Time per epoch vs time to completion** We chose time per epoch as it is easier to measure directly than time to completion, as completion time varies depending on when our convergence condition is triggered (see appendix D.3. for details). Both KIP and RFAD converge in a similar number of iterations (3k - 15k iterations, see section 4.1), meaning that the time savings per iteration are a good proxy for the total time savings. Table 4 in appendix E has exact details.
> > >
> > > We hope that this helps clarify our time complexity argument. Indeed, this argument is much more nuanced than what we presented in the paper, and we will add this more careful discussion into the appendix in the camera-ready version.

---

### Meta-Review · Area_Chair_gpPy · 2022-08-30

**Recommendation:** Accept
**Confidence:** Certain

**Metareview:**

The paper provides a novel and practical algorithm for dataset distillation. The paper is clearly written and the reviewers felt that this is a nice contribution to the field. The results demonstrate a clear improvement over state of the art and the experiments are sound. The reviewers raised concerns about limited novelty of the paper but, after a fruitful discussions, agreed that the paper should be accepted.

**Award:**

No

---

### Decision · Program_Chairs · 2022-09-14

Accept